# Dead Bodybuilders Speaking from the Heart: An Analysis of Autopsy Reports of Bodybuilders That Died Prematurely

**DOI:** 10.3390/jfmk7040105

**Published:** 2022-11-24

**Authors:** Guillermo Escalante, Dillon Darrow, V. N. Pradeep Ambati, Daniel L. Gwartney, Rick Collins

**Affiliations:** 1Department of Kinesiology, California State University San Bernardino, San Bernardino, CA 92407, USA; 2Independent Researcher, Columbia, MO 65201, USA; 3Collin, Gann, McCloskey & Barry PLLC, Attorneys at Law, Mineola, NY 11501, USA

**Keywords:** ventricular hypertrophy, anabolic-androgenic steroids, atherosclerosis, hypertension, cardiomegaly

## Abstract

This study analyzed publicly available autopsy reports of male bodybuilders under the age of 50 who reportedly died from cardiovascular-related events. A general Google search with the terms “dead bodybuilders” was performed on 10 February 2022. Six reports were available for review and analysis. Bodybuilders had the following means: age (±SD) = 36 ± 7.1 years; height 1.82 ± 0.02 m; weight = 103.8 ± 5.3 kg; weight of heart = 575 ± 134.4 g; and left ventricular myocardium thickness (*n* = 3) = 16.3 ± 3.5 mm. The bodybuilders analyzed had a mean heart weight that is 73.7% heavier than the reference man (575 g vs. 332 g). Similarly, 100% of the autopsies reported left ventricular myocardium thickness of 16.3 ± 3.5 mm; this is 125% thicker than normative data for men. While abuse of AASs for prolonged periods of time may contribute to some of the cardiac abnormalities present in these bodybuilders, it should be noted that cardiac hypertrophy, including left ventricular hypertrophy, has also been reported in drug-free strength athletes. Each autopsy report included cardiovascular abnormalities within the cause of death. Association does not mean causation, but nonetheless bodybuilders should be aware of potential contributing cardiovascular risks with AAS abuse.

## 1. Introduction

Use of performance-enhancement drugs (PEDs), in particular androgenic-anabolic steroids (AASs), has been reported by non-competitive individuals as well as athletes in sports such as cycling, track and field, baseball, football, weightlifting, soccer, bodybuilding, and many others [1,2,3,4,5]. Specifically in bodybuilding, a recent study reported that 53.6% of respondents participating in competitions not sanctioned as “natural” and 2.4% of respondents competing in “natural” bodybuilding events utilized PEDs inclusive of testosterone and other AASs, such as trenbolone acetate, oxandrolone, stanozolol, and drostanolone enanthate [6]. Other studies have also reported that the prevalence of AAS use among bodybuilders ranges from 20.6% to 63.5% [7,8,9,10,11].

In a study on the nutrition, training, supplementation, and pharmacological practices of amateur physique competitors in southern California currently under review from one investigator in our group (Escalante), ~48% of males reported using PEDs inclusive of AASs. Since this study was conducted in amateur competitors at regional level/non-tested physique competitions, it is likely that the prevalence of AAS use among competitors is significantly higher at more competitive non-tested events, such as national level pro-qualifying events or professional level competitions where more muscularity is required to be successful.

Gaining large quantities of muscle mass, as seen among top-level professionals in non-tested federations, requires years of consistent weight training fueled with large quantities of food consumed in regimented fashion along with nutritional supplements [12,13,14,15,16]. For many, it may also require the use of PEDs, including supraphysiological doses of testosterone, other AASs, and other anabolic compounds [6,17]. While AASs have many medical uses and implications, long-term supraphysiological use of AASs has been associated with negative health effects, especially to the cardiovascular system [18,19,20,21]. Common cardiovascular issues reported with AAS abuse include elevated levels of low-density lipoprotein (LDL), low levels of high-density lipoprotein (LDL), elevated systolic and diastolic blood pressure, and left ventricular hypertrophy [18,19,20,21].

The recent deaths or near deaths of male and female bodybuilders has sparked significant concern in the bodybuilding community and beyond. A competitive 46-year-old bodybuilder recently discussed how AAS use likely contributed to the heart attack he survived in 2021 [22]. In another article, the author reported the death of 15 top level professional and high-ranking amateur bodybuilders in the year 2021 of which 80% were ≤ 50 years old [23]. Although the author accidentally misreported the death of one of the bodybuilders, which has since been corrected, as he is still alive, the others have been confirmed. Since several of the reported deaths or near deaths among bodybuilders under the age of 50 in 2021 were reported to be related to cardiovascular events according to various sources [22,23,24], the primary purpose of this investigation was to analyze publicly available autopsy reports of male bodybuilders who were reported to have died from cardiovascular-related events before the age of 50. Secondly, we reported the official coroners’ findings inclusive of reported cause of death, height, weight, information about the bodybuilder’s cardiovascular system, and other relevant information, such as pertinent clinical history/medications and toxicology reports, when available.

## 2. Materials and Methods

A general Google search with the term “dead bodybuilders” was performed on 10 February 2022 that yielded results to eighteen websites in the first two pages of the search. After eliminating duplicate information and YouTube videos, the seven sources found in Table 1 were used as starting points for the search. Next, each source was visited individually to identify male bodybuilders who died before the age of 50 for any cause in the last 12 years; a total of 45 bodybuilders were identified. This was further filtered by those who were reported to have died due to a heart attack, heart failure, other cardiovascular events (e.g., stroke, embolism, etc.), natural causes, or unknown causes and where the location (state) and date of death were available. Bodybuilders who passed away outside of the United States were excluded from the analysis, as it was not possible to obtain autopsy reports outside of the USA. In total, 14 bodybuilders were identified, and members of our research team contacted each individual county coroner’s office to request full autopsy reports for review (Table 2). A total of seven reports were available for review and analysis, but one was excluded because the official cause of death was not a cardiac-related event; thus, we analyzed six autopsy reports. Since all information collected and analyzed was publicly available, no institutional review board was necessary for this study. However, to maintain some level of anonymity for the deceased bodybuilders, the names of the bodybuilders in this report are not shared. The data for this study is reported descriptively or as percentages.

## 3. Results

The demographics of the deceased bodybuilders where autopsies were available are reported in Table 3. Figure 1 shows the weight of the heart of each bodybuilder and compares it to the weight of the heart for the reference man [30]. The mean left ventricular myocardial thickness for the reported bodybuilders (*n* = 3) was 16.3 ± 3.5 mm. Prevalence of left ventricular hypertrophy was 100%, chamber dilation was 33%, positive use of anabolic steroids was 67%, and any drugs found in their possession was 100%. In the five bodybuilders where atherosclerosis was reported and a toxicology report was performed, 80% had atherosclerosis and 60% had a positive toxicology report for illicit drugs. Causes of death reported by the coroners included heart disease, steroid-induced cardiomyopathy, sudden cardiac dysrhythmia, and left ventricular hypertrophy. Summaries of the coroner report findings are shown in Table 4.

## 4. Discussion

The data in this study show that the deceased bodybuilders had a mean heart weight that is 73.7% heavier than the reference man [30] (575 g vs. 332 g, respectively). Similarly, the autopsies in this investigation that measured left ventricular myocardial thickness (*n* = 3) reported a mean thickness of 16.3 ± 3.5 mm; this is 125% thicker than normative data for men [31]. Additionally, chamber dilation was present in 33% of the bodybuilders, and the prevalence of atherosclerosis was 80% in the five autopsies where it was reported. The coroners’ reported causes of death for the bodybuilders included heart disease, steroid-induced cardiomyopathy, sudden cardiac dysrhythmia, and left ventricular hypertrophy. Abuse of AASs for prolonged periods of time may have contributed to some of the cardiac abnormalities present in these bodybuilders; however, more research is required before being able to draw more definitive conclusions.

Some research does suggest that long-term AAS abuse may lead to cardiovascular complications. For example, Achar and colleagues reviewed 49 studies that described 1467 athletes that used AASs and reported that elevated levels of low-density lipoprotein (LDL), low levels of high-density lipoprotein (HDL), elevated systolic and diastolic blood pressure, and left ventricular hypertrophy were associated with AAS abuse [18]. In another study, Smit et al. investigated the effects of AAS use on left ventricular hypertrophy and cardiac dysfunction prospectively in 31 men that had self-selected to start an AAS cycle [21]. Investigators reported that after a median AAS cycle of 16 weeks, 3D left ventricular ejection fraction declined, ratio of the early to late ventricular filling velocities decreased, 3D left atrial volume increased, and left ventricular mass increased [21]. While these values returned back to baseline after ~8 months of participants stopping their cycle, median dose was supraphysiological at ~904 mg per week during the 16-week cycle; this may be lower than that used by high level enhanced bodybuilders that potentially run longer cycles of 5–20+ consecutive years with few breaks, if any, in between cycles.

In another study, Barbosa Neto et al. examined the cardiovascular and autonomic abnormalities in bodybuilders actively using AASs as compared to lifetime natural bodybuilders and healthy sedentary controls [19]. In this study, the authors reported that AAS users had higher mean blood pressure, higher sympathetic modulation, lower parasympathetic modulation, and increased thickness in the interventricular septum, left ventricle posterior wall, and relative diastolic wall as compared to lifetime natural bodybuilders and healthy controls [19]. Prior to this investigation, Baggish et al. examined the long-term effects of AAS use on left ventricular dysfunction by performing 2D tissue-Doppler and speckle-tracking echocardiography to assess left ventricular (LV) ejection fraction, LV systolic strain, and conventional indices of diastolic function in long-term AAS users as compared to non-users that were matched for age, weight lifting experience, and weight training intensity [20]. The investigators reported that AAS users had significantly lower LV ejection fraction, longitudinal strain, and radial strain, and that ~83% of AAS users showed LV ejection fractions below the accepted limit of normal (≥55%) [20]. Interested readers can also refer to a recent position statement by the European Association of Preventative Cardiology on the cardiovascular effects of doping substances to see other studies that have suggested that AAS abusers are at increased risk for cardiomyopathy and LV dysfunction [5].

Obesity is frequently associated as a contributing factor of heart disease; however, angiographic and post-mortem studies have demonstrated little or no correlation of total fat mass and coronary atherosclerosis except in those with abdominal obesity [32]. The bodybuilders in this investigation had a mean body mass index (BMI) of 31.6 ± 2.3 kg/m^2^. While this BMI categorizes the bodybuilders as obese, these bodybuilders likely did not carry large quantities of fat mass or abdominal adiposity. Specifically, two of the deceased bodybuilders from this investigation were less than 2 weeks out from competing, and it is well known that bodybuilders drop to low single digit levels of body fat percentage when they are ready for competition [33,34]. Furthermore, research suggests that there is a weak correlation between BMI and body fat percentage in individuals who engage in vigorous resistance training [35]. As such, the high BMI in these bodybuilders is likely related to the large quantities of fat-free mass (as opposed to fat mass) that they carry; this can be quantified using the fat-free mass index (FFMI). The FFMI, which is derived from an individual’s fat-free body mass in kilograms divided by their height in meters squared, has been used as a rudimentary tool to help identify AAS users from non-users [36]. Based on some of the original data of the FFMI on AAS users and non-users, an FFMI ≥ 25 kg/m^2^ has been suggested as an initial measure to screen for possible AAS use among athletes [36]. Although FFMI has inherent limitations, an FFMI index much higher than 25 kg/m^2^ is unlikely to be achieved by most natural individuals, yet most (if not all) professional bodybuilders in untested federations have a FFMI > 25 kg/m^2^. For example, the two bodybuilders in this investigation that were getting ready to compete had a mean body weight of 93.2 kg and mean height of 1.69 m. Assuming a conservative body fat level of 5% for these bodybuilders prior to competing, their estimated FFMI is 31.0 kg/m^2^. Since this is significantly higher than the FFMI cut-off for non-AAS users, it supports the notion that supraphysiological doses of AASs likely played a role in their muscular development and thus may have contributed to some of their cardiovascular complications.

While this study provides additional preliminary data that AAS abuse may contribute to cardiovascular complications that may lead to death, it has some inherent limitations. One major limitation is the quality and thoroughness of the autopsy reports. The reports vary significantly in length and detail; thus, some are inherently more thorough than others. The lack of consistency in which autopsies are conducted when young athletes die if there is suspicion of AAS abuse makes it difficult to draw more direct cause and effect conclusions. Indeed, Esposito et al. describe a thorough investigative protocol to conduct autopsies for cases of suspected death related to AAS abuse, and the authors encourage clinicians to utilize a more accurate and methodologic approach when doing autopsies [37]. A second limitation is the small sample size of six autopsies. Although we tried to obtain more reports, these were the only ones we were able to gather. As such, it is difficult to draw definitive conclusions from such a small sample size. Another limitation is that the exact dosages and length of exposure to AASs in this group remain unknown. Additionally, since AASs are not legally prescribed at supraphysiological doses for the purpose of enhancing performance, the quantity, quality, and purity of the compounds that may have come from underground sources that may have been used by the competitors is also unknown. Furthermore, the use of other drugs that are not AASs, such as insulin, human growth hormone, or recreational drugs, and/or genetic predispositions to cardiovascular problems, further complicate the interpretations of the findings of this study. Importantly, it should also be noted that cardiac hypertrophy, which was found in all of these bodybuilders, has also been reported in drug-free weightlifters [38]. Thus, our comparison of the hearts of the dead bodybuilders to the reference man as opposed to other drug-free bodybuilders or weightlifters should be interpreted with caution. Lastly, association does not mean causation, especially since there are confounding variables involved that may have contributed to the premature deaths of these bodybuilders. Future research is required to further clarify the potential risks of cardiovascular complications with continual supraphysiological abuse of AASs in bodybuilders.

## 5. Conclusions

Abuse of AASs for prolonged periods of time may have contributed to some of the cardiac abnormalities present in these bodybuilders; however, more research is required before being able to draw more definitive conclusions. Cardiomegaly, left ventricular hypertrophy, and/or atherosclerosis was present in the bodybuilders examined. However, association does not mean causation, especially since there are confounding variables involved that may have contributed to the premature deaths of these bodybuilders. Future research is required to further clarify the potential risks of cardiovascular complications with continual supraphysiological abuse of AASs. Furthermore, future autopsies for cases of suspected death related to AAS abuse would benefit from a more accurate and methodologic approach when doing the procedures.

## Figures and Tables

**Figure 1 jfmk-07-00105-f001:**
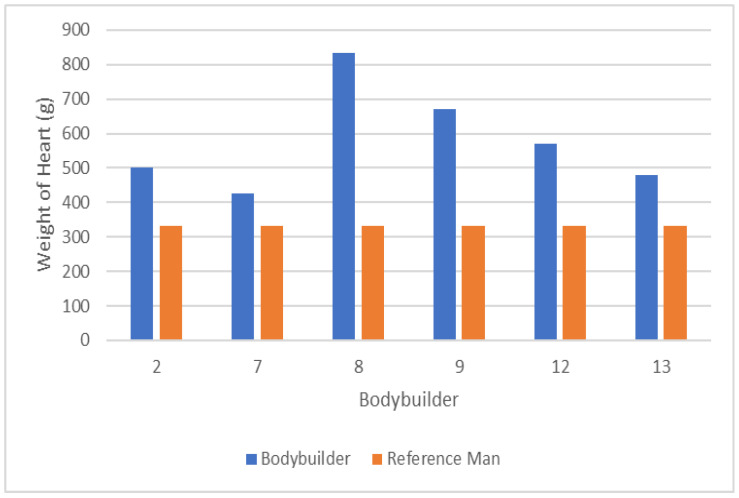
Mean weight of hearts versus reference man.

**Table 1 jfmk-07-00105-t001:** Search results for dead bodybuilders.

Website (Accessed on 11 February 2022)	Reference
www.heartwise.com	[23]
www.the-sun.com	[24]
www.evolutionofbodybuilding.net	[25]
www.fitnessvolt.com	[26]
www.playcrazygame.com	[27]
www.t-nation.com	[28]
www.thebarbell.com	[29]

**Table 2 jfmk-07-00105-t002:** Reported deceased bodybuilders (age < 50 years) in the USA from cardiovascular events, natural causes, or unknown causes between 2010–2022.

Bodybuilder	Unofficial Reported Cause of Death	Birth Year	Date of Death	Place of Death
1	Myocardial infarction	1975	11/6/2021	Nevada
2	Sudden cardiac arrest	1984	10/6/2021	Florida
3	Pulmonary embolism/ Myocardial infarction	1972	8/8/2021	Ohio
4	Natural causes	1983	5/21/2020	Alabama
5	Unknown	1985	6/3/2020	New Jersey
6	Heart failure	1985	5/30/2019	Oregon
7	Natural via cardiomyopathy	1989	8/22/2017	California
8	Myocardial infarction	1991	8/22/2017	Florida
9	Enlarged organs	1970	8/25/2017	Florida
10	Undisclosed	1970	1/24/2015	California
11	Myocardial infarction/heart failure	1965	8/16/2014	California
12	Myocardial infarction	1972	2/20/2013	Georgia
13	Accidental drowning/ Myocardial infarction	1973	9/11/2011	Texas
14	Myocardial infarction	1966	2/19/2011	Nevada
Sources: [23,24,25,26,27,28,29]				

**Table 3 jfmk-07-00105-t003:** Bodybuilder demographics.

Bodybuilder	Age (Years)	Height (m)	Weight (kg)	Body Mass Index kg/m^2^
2	37	1.68	93.2	33
7	30	1.7	93.2	32.2
8	26	1.83	120	35.8
9	46	1.83	100	29.9
12	40	1.83	127.3	38
13	37	1.8	107.5	33.2
Mean ± SD	36 ± 7.1	1.82 ± 0.02	103.8 ± 5.3	31.6 ± 2.3

**Table 4 jfmk-07-00105-t004:** Coroner report summary findings.

Bodybuilder	Coroner Summary of Findings
2	Sudden cardiac dysrhythmia due to hypertensive cardiovascular disease. Cardiomegaly with ventricular hypertrophy. Anabolic steroid use noted as a contributory factor. The coronary arteries are patent and have no arteriosclerosis. The atrial and ventricular chambers are not dilated. The papillary muscles and chordae tendineae are thickened.
7	The cause of death is steroid-induced cardiomyopathy. There is segmental atherosclerosis with up to 50% stenosis of the major arteries. Left ventricular dilation is present.
8	Bodybuilder with a history of dyslipidemia (low HDL, elevated LDL), elevated aminotransferases, and a chronic cough with some episodes of associated shortness of breath. Family history of early-onset atherosclerosis and hypertension. He collapsed during a period of ongoing respiratory symptoms. Follow-up evaluation showed abnormal electrocardiogram that met criteria for left ventricular hypertrophy. Severe cardiomegaly with concentric left ventricular hypertrophy. Coronary artery atherosclerosis.
9	Cardiomegaly with mild coronary atherosclerosis. The heart was enlarged with the myocardium of the left and right ventricles hypertrophied. The coronary arteries were normally developed and had mild atherosclerotic disease. The chambers were not dilated and the atrial and ventricular septae were intact.
12	The deceased died as a result of cardiac dysrhythmia associated with cardiomegaly and left ventricular hypertrophy.
13	Hypertensive and arteriosclerotic cardiovascular disease. Cardiomegaly with left ventricular hypertrophy. Coronary artery disease. Remote scar of myocardium. Generalized atherosclerosis. There are hypertensive changes of myocytes. A section of coronary artery reveals complicated atherosclerotic plaque with recanalization. The heart has four chamber dilation.

## Data Availability

The data supporting the conclusions of this article will be made available by the corresponding author, without undue reservation.

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
