# Peer review of "Dead Bodybuilders Speaking from the Heart: An Analysis of Autopsy Reports of Bodybuilders That Died Prematurely"

_jfmk, 2022, doi:10.3390/jfmk7040105_

Round 1

Reviewer 1 Report

Congratulations on the idea and on bringing up an important and overlooked topic

In my opinion, the strengths of the work are: Bringing up an important topic due to the significant percentage of people abusing SSA even in amateur heavy lifting. Great idea for material gathering. Conclusions carefully formulated, adequate to the results

Weak points Overall, a very small group was assessed based on autopsies performed according to different, possibly incomparable standards. The authors do not refer to the proper criterion for the diagnosis of cardiomyopathy, i.e. septal hypertrophy. Instead, they serve you with all your heart.The group comparison methodology is also weak: dead bodybuilders versus reference mean is less accurate than for example dead and alive bodybuilders. From the results, it cannot be concluded that the SSAs caused the cardiomyopathy or were in any way related to the immediate cause of death. It is unknown whether the deceased did not have cardiomyopathy originally. The authors also forgot that it is not only AAS that are responsible for myocardial hypertrophy. Increasingly popular and dangerous is the use of growth hormone. The authors did not comment on this at all. Consider inclusion od more limitation (mentioned above) or maybe some change of results presentation.

General Although the topic is interesting, with the methodology used, the work does not bring new knowledge that would be of reference to other studies.

Author Response

Thank you very much for your excellent comments. We feel our paper is improved with your input. The revised manuscript has the changes highlighted. 

We agree that we had a very small group was assessed based on autopsies performed according to different, possibly incomparable standards. We added this limitation to our discussion. We also have a section in the limitations that states this "One major limitation is the quality and thoroughness of the autopsy reports. The reports vary significantly in length and detail; thus, some are inherently more thorough than others. The lack of consistency in which autopsies are conducted when young athletes die if there is suspicion of AAS abuse makes it difficult to draw more direct cause and effect conclusions. Indeed, Esposito et al. describe a thorough investigative protocol to conduct autopsies for cases of suspected death related to AAS abuse and the authors encourage clinicians to utilize a more accurate and methodologic approach when doing autopsies [37]." 

The authors do not refer to the proper criterion for the diagnosis of cardiomyopathy, i.e. septal hypertrophy. Instead, they serve you with all your heart. Thank you, You are right that hypertrophic cardiomyopathy was originally defined as septal hypertrophy. However, to our knowledge, the the definition of hypertrophic cardiomyopathy has been broadened  because segmental hypertrophy can occur in any segment of the ventricle, not just the septum. 

The group comparison methodology is also weak: dead bodybuilders versus reference mean is less accurate than for example dead and alive bodybuilders. From the results, it cannot be concluded that the SSAs caused the cardiomyopathy or were in any way related to the immediate cause of death. It is unknown whether the deceased did not have cardiomyopathy originally. The authors also forgot that it is not only AAS that are responsible for myocardial hypertrophy. Increasingly popular and dangerous is the use of growth hormone. The authors did not comment on this at all. Consider inclusion of more limitation (mentioned above) or maybe some change of results presentation. Thank you for this. we added more limitations to our discussion to highlight this.    

Reviewer 2 Report

The manuscript is well written, the topic is of scientific interest and the outcomes may contribute in studying the cardiovascular side effects of AAS abuse in athletes and particularly in bodybuilders.

Specific Comments

1.     Line 51. Add reference

2.     Lines 168-172. It would add value to your discussion here if you had found actually measured values of Body Fat % of the victims (if recorded any) and not to make assumptions even with a solid rationale as you do.

3.     The study has some major limitations that they you already mentioned in the text i.e. the dosage scheme and duration of AAS use, the ingredients of the used supplements or a genetic post-mortem analysis, these would significantly improve the interpretation of the results. Furthermore, the size of the final sample of bodybuilders is rather small.

4.  Consider also using this recent position statement reference “Adami, P.E., Koutlianos, N., Baggish, A., et al. (2022). Cardiovascular effects of doping substances, commonly prescribed medications and ergogenic aids in relation to sports: a position statement of the sport cardiology and exercise nucleus of the European Association of Preventive Cardiology. Eur J Prev Cardiol 29: 559–575, 2022. doi:10.1093/eurjpc/zwab198.”

Author Response

Thank you for reviewing our paper. Your suggestions have improved the quality of our paper. All modifications to the paper have been highlighted in the revised draft. Please see our responses to your comments below:

  1. Line 51. Add reference. Reference was provided in line 51 (Reference 22). I hope this is what you meant.
  2. Lines 168-172. It would add value to your discussion here if you had found actually measured values of Body Fat % of the victims (if recorded any) and not to make assumptions even with a solid rationale as you do. We agree. Unfortunately this data was not available.
  3. The study has some major limitations that they you already mentioned in the text i.e. the dosage scheme and duration of AAS use, the ingredients of the used supplements or a genetic post-mortem analysis, these would significantly improve the interpretation of the results. Furthermore, the size of the final sample of bodybuilders is rather small. We agree. Thank you. We included all of these limitations in the discussion. We also added a few other limitation in the discussion.
  4. Consider also using this recent position statement reference “Adami, P.E., Koutlianos, N., Baggish, A., et al. (2022). Cardiovascular effects of doping substances, commonly prescribed medications and ergogenic aids in relation to sports: a position statement of the sport cardiology and exercise nucleus of the European Association of Preventive Cardiology. Eur J Prev Cardiol 29: 559–575, 2022. doi:10.1093/eurjpc/zwab198.” Great. Thank you. We have added this reference to the intro and discussion. This is indeed a great paper.